# Nickel-Aluminum Thermal Spray Coatings as Adhesion Promoter and Susceptor for Inductively Joined Polymer-Metal Hybrids

**DOI:** 10.3390/polym13081320

**Published:** 2021-04-17

**Authors:** Erik Saborowski, Axel Dittes, Thomas Lindner, Thomas Lampke

**Affiliations:** Materials and Surface Engineering Group, Faculty of Mechanical Engineering, Chemnitz University of Technology, Erfenschlager Straße 73, D-09125 Chemnitz, Germany; axel.dittes@mb.tu-chemnitz.de (A.D.); th.lindner@mb.tu-chemnitz.de (T.L.); thomas.lampke@mb.tu-chemnitz.de (T.L.)

**Keywords:** mechanical interlocking, polymer-metal hybrid, fiber-reinforced polymer, inductive joining, lap shear test, hygrothermal cyclic loading, thermal spraying

## Abstract

Hybrid joints of metal- and fiber-reinforced-polymer offer great potential for lightweight applications. Thereby, a fast and reliable joining process is mandatory for mass-production applications. To this end, this study assesses inductive spot-joining in combination with prior thermal spray coating of the metal adherent. A nickel–aluminum 95/5 coating was applied to achieve high adhesion through mechanical interlocking and to act as susceptor for the inductive joining process. The joint strength was assessed with lap shear specimens consisting of EN AW-6082 aluminum alloy and glass fiber reinforced polyamide 6 or polypropylene, respectively. The joints were further investigated in terms of heating time and hygrothermal cyclic loading. The results showed that significant time savings for the joining process as well as strong adhesion were achieved due to the coating. Moreover, the high strengths were even preserved under hygrothermal cyclic loading.

## 1. Introduction

Thermoplastic polymer-metal hybrids offer great potential for automotive and aviation applications due to their high strength/stiffness-to-weight ratio. In contrast to hybrid structures containing thermoset polymers, no time-consuming curing process is necessary. Thereby, key challenges are the development of a well adhering interface as well as a fast joining technique. Common industrial joining methods are e.g., adhesive bonding [1], riveting [2], or clinching [3]. Another promising approach is thermal joining. Hereby, the polymer itself serves as hot-melt adhesive as it infiltrates and interlocks with the roughness features of the metallic surface. Metal and polymer are put together under pressure, while the contact area between both materials is heated until the polymer melts. The necessary thermal energy can e.g., be generated by direct laser heating [4,5], indirect laser heating [6,7,8,9], ultrasonic oscillations [10,11,12], friction [13,14], heat conduction [15,16,17] or induction [8,18,19,20]. Depending on the respective design, all methods are able to generate the necessary heat within the low-second or even sub-second range. Advantages of inductive heat generation are joining without damaging the metal surface like friction based or ultrasonic methods, easy integration into joining tools as well as low investment costs [21]. The susceptor that is needed for absorbing the electromagnetic energy can be integrated into the polymer [18,19,22,23]. Furthermore, the metal part itself can serve as susceptor [8,20].

Metal surface topography has a major impact on the achievable strength of the joint. The most widely used structuring method is mechanical blasting [6,17,24] since it offers acceptable bonding strength and can easily be integrated into industrial applications. However, high residual stresses arise in the near-surface area, leading to considerable deformation of the metallic adherent. Laser structuring processes offer the highest possible bonding strength [7,17,25,26]. However, the processing times are very high. Thermal spray coatings provide a rough and undercut surface. Hence, they can be used as adhesion promoter for polymer-metal hybrids, reaching an interlaminar strength higher than mechanically blasted metallic surfaces [17,26,27]. When the coating is applied without prior activation of the substrate, residual stresses due to mechanical blasting can be avoided. Although thermal spray coatings also exhibit residual stresses, their distribution is limited to the thickness of the relatively thin coatings, whereas mechanical blasting usually has a much higher penetration depth [28]. Consequently, excessive deformation of thin substrates is avoided. Another advantage of thermally sprayed coatings is the ability to use a wide range of materials. By using a ferromagnetic metal with high permeability, the coating can also serve as susceptor in addition to its purpose as adhesion promoter since joule heating caused by induced eddy currents as well as hysteresis losses lead to an acceleration of the inductive heating process [21].

The utilization of a coating as both susceptor and adhesion promoter enables a fast and reliable process chain for producing strong joints between metal and thermoplastic polymer. To this end, this contribution investigates self-adhering, ferromagnetic nickel–aluminum 95/5 coatings for inductive joining of EN AW-6082 aluminum alloy to glass fiber reinforced polypropylene and polyamide 6. A parameter study for coating application by wire arc spraying is conducted to achieve the highest possible joint strength. The joined specimens are strength-tested by lap shear tests. Furthermore, the coatings are investigated in terms of their surface roughness to assess the relation between surface structure and joint strength. The heating rates, depending on coating thickness and induction coil distance, are investigated by infrared camera images. Additionally, the joined specimens are investigated for their resistance against hygrothermal cyclic loading by climate tests to investigate how physical and chemical aging of the polymer affect the joint strength. Finally, the fracture surfaces are analyzed to determine the occurring failure mechanisms.

## 2. Materials and Methods

Metal sheets made up from 100 × 25 × 3 mm^3^ EN AW-6082 T6 aluminum alloy (Hans-Erich Gemmel & Co. GmbH, Berlin, Germany) as well as polymer sheets made up from 80 × 25 × 1 mm^3^ Tepex^®^ dynalite 102-RG600 (LANXESS Deutschland GmbH, Cologne, Germany) glass fiber reinforced polyamide 6 (PA6GF47BD) and Tepex^®^ dynalite 104-RG600 glass fiber reinforced polypropylene (PPGF47BD) were used. Table 1 shows the supplier specifications of the used material. The lap shear tests were conducted in accordance to DIN EN 1465 for adhesively bonded assemblies. The utilized specimen geometry is illustrated in Figure 1a.

Ø1.6 mm DURMAT^®^ AS-756 Nickel–Aluminum 95/5 (NiAl5) cored wire (DURUM Verschleißschutz GmbH, Willich, Germany) was utilized for coating deposition on the aluminum substrate. The coatings were applied by wire arc spraying unit VISU ARC 350 with Schub 5 spraying gun (Oerlikon Metco, Wohlen, Switzerland). For determination of a suitable combination of current and voltage for the wire arc spraying process, a parameter study with the values V1–V6, shown in Table 2, was conducted. The best parameter set in terms of lap shear strength was further utilized for assessing the influence of the coating thickness on the inductive spot-joining process. Therefore, a number of 1–8 layers (1L–8L) was investigated.

Prior to the joining process, the aluminum and polymer sheets were ultrasonically cleaned in ethanol. Additionally, the PA6GF47BD was dried at 70 °C for three days to ensure a low moisture content and avoid bubble formation during the joining process. The lap shear specimens were manufactured with the joining device shown in Figure 2a. Metal and polymer were pressed together with a spring-loaded stamp with Ø12 mm, using a constant force of 125 N. Since the aluminum sheet as well as the coating can serve as susceptor, both variants were examined. In the initial experiments for determining the best coating parameters, induction coil and loading stamp were placed above the aluminum sheet. Thereby, an even stress distribution and wetting was achieved in the joining zone. Hence, the bearable lap shear strength can be calculated. Deviating from DIN EN 1465, a shortened overlap length of 5 mm was used to limit the maximum force during the lap shear test and thus avoid fracture of the polymer sheet. In the following experiments investigating spot-joining, the induction coil was placed above the polymer and the coating was used as susceptor. Since the polymer sheets were not stiff enough for distributing the stress evenly during the joining process, no complete wetting of the overlapping area was realized. Hence, only the breaking force is given. An increased overlapping length of 15 mm was used to ensure full support of the spring-loaded stamp. Figure 1b shows a spot-joined specimen.

The inductive heating was performed with a HFL 02/5 5 kW high frequency converter (Frisch GmbH, Pforzheim, Germany). The utilized induction coil had an outer diameter of 22 mm, an inner diameter of 18 mm and a wall thickness of 1 mm. The coil was placed 0.5 mm above the aluminum and directly on the polymer, respectively. A power setting of 75% was used for joining the specimens. The power was active until the temperature in the joining zone reached 200 °C for the PPGF47BD specimens and 260 °C for the PA6GF47BD specimens. The temperature in the joining zone was observed with a HH507 thermometer (OMEGA Engineering, Deckenpfronn, Germany). After deactivating the high frequency converter, the joined specimens cooled down for 15 s before removal. Prior strength testing, the PA6GF47BD specimens were conditioned according to DIN EN ISO 1110 for three days to ensure similar moisture content and mechanical behavior of the polyamide 6. The lap shear tests were carried out with an Allround-Line 20 kN testing machine (ZwickRoell GmbH & Co. KG, Ulm, Germany), using a crosshead speed of 1 mm/min. Three lap shear specimens were tested for each parameter set.

The influence of the coating thickness on the heating rate was determined with a VarioCAM^®^ HD head 900 infrared camera (JENOPTIK Advanced Systems GmbH, Jena, Germany). The coil was placed 1, 3 and 5 mm above the coating and the high frequency converter was activated at 50% power setting for slightly over 5 s. The resulting temperature curve represents the time-dependent mean value of a circle with Ø10 mm in the center of the coil (Figure 2b). The recorded data was evaluated with IRBIS 3.1 plus software (InfraTec GmbH, Dresden, Germany). For the oxidized nickel surface, an emission coefficient of ε = 0.37 was assumed. The resulting ΔT represents the temperature difference between start of the heating process and 5 s heating time.

The hygrothermal cyclic loading test was carried out in accordance to test standard BMW PR 308.2 described by [25] with a VC 4018 climate chamber (Vötsch Industrietechnik GmbH, Reiskirchen-Lindenstruth, Germany). The test cycle switched between 90 °C at 80% humidity and −30 °C at ambient humidity. The dwell time at the maximum and minimum temperatures was 4 h. The switching time in-between was 2 h. Overall, 20 cycles were performed, resulting in a total test time of 240 h.

The cross-sectional images were recorded with GX51 optical microscope (Olympus Europe SE & Co. KG, Hamburg, Germany). The coating thicknesses were determined with ImageJ 1.52a image evaluation software [29] from the mean value of 10 measurements between substrate and top of the coating each. The fractured surfaces were evaluated with MVX10 optical microscope (Olympus Europe SE & Co. KG, Hamburg, Germany).

The roughness measurements were carried out in accordance to DIN EN ISO 4287, using a Hommel-Etamic^®^ T8000 stylus profiler (JENOPTIK AG, Jena, Germany). Five measurements with an evaluation length of 12.5 mm were recorded and evaluated for coating parameter sets V1–V6. Thereby, a stylus tip wit 2 µm radius and 60° tip angle was used for capturing the highest possible amount of profile details. Average maximum profile height *Rz*, arithmetical mean height *Ra* and root mean square slope *R*Δ*q* were evaluated. Especially the slope has shown good accordance in past studies since it is directly related to the density and the aspect ratio of the surface roughness features [17,30].

## 3. Results and Discussion

Figure 3 illustrates the thinnest (V4: 90.8 ± 38.7 µm) as well as the thickest (V3: 180.3 ± 28.7 µm) coating within the investigated spraying parameter range. In all cases, a complete bond between coating and substrate, as well as between the individual coating layers, was achieved. The electric spraying parameters affect the amount of molten wire during the spraying process and therefore the thickness of the coating (Figure 4a). However, the coating surface structure and morphology are only slightly affected. The roughness measurements revealed rather small differences, especially for the most meaningful RΔq value. The lowest values were measured for V4 (Rz = 104 ± 8 µm, Ra = 16.0 ± 1.5 µm, RΔq = 0.855 ± 0.031), the highest values were measured for V3 (Rz = 131 ± 6 µm, Ra = 21.6 ± 1.8 µm, RΔq = 0.943 ± 0.035). Since the strength between coating and polymer mainly depends on the surface structure, the achieved lap shear strength in connection with PA6GF47BD varied only within a small range of 19.7 ± 0.4 MPa and 21.0 ± 0.7 MPa (Figure 4b). This is roughly in the range of 24 MPa that were reported in a recent study for the shear strength between fiber-reinforced polyamide 6 and NiAl5 coatings [27].

The heating experiments revealed a strong influence of the coating thickness on the heating rate. Figure 5a illustrates the temperature increase ΔT after 5 s of heating at 50% power setting in dependence of the coil distance and the coating thickness. A thick coating enables more absorption of electromagnetic energy then a thin coating and thus leads to higher heating rates. Moreover, increasing the coil distance drastically decreases the heating rate. This is also reported by [23] as a result of the decrease in magnetic field intensity with increasing coil distance. Figure 5b shows exemplary temperature curves for coating parameter set V2. Especially when the coil distance increases, a thick coating still enables reasonable heating rates, whereas only a negligible temperature increase can be achieved without coating (coil distance 5 mm, no coating: ΔT = 10.5 K, 180.3 µm coating: ΔT = 73.5 K).

The spraying parameter set V2 was chosen for further investigations as it offered the highest lap shear strength. A number of one to eight layers were investigated to determine the influence of the coating thickness on the actual spot-joining process. As expected, the number of layers is directly connected to coating thickness (Figure 6a). The necessary heating time shown in Figure 6b decreased with increasing number of layers (1L: 14.92 ± 1.87 s, 6L: 7.64 ± 1.51 s). However, a further increase in the number of layers from six to eight did not lead to a further reduction in the heating time (8L: 8.08 ± 0.53 s). This is due to the complete absorption of the magnetic field by the layer above a certain thickness. A further increase in layer thickness therefore has no positive effect on the inductive heating rate. The breaking force shown in Figure 6c varied within a small range of 4158 ± 203 N to 4534 ± 369 N. Consequently, no significant influence of the layer number on the joint strength can be detected.

Since variant 6L provided the fastest heating rate as well as the highest breaking force, it was further investigated in connection with PPGF47BD. The breaking force achieved with PPGF47BD (3140 ± 172 N) is considerably lower than the breaking force achieved with PA6GF47BD (4534 ± 369 N) due to the lower strength of polypropylene matrix material in comparison to polyamide 6 matrix material. In contrast to the results reported by [25] for the 90 °C/−30 °C climate test, there was no significant decrease in breaking force for both polymer materials (PA6GF47BD: −4.5%, PPGF47BD: −0.8%). This is possibly because [25] used unreinforced and short-fiber reinforced polymer, which is subject to greater residual stresses and thus higher damage of the interface due to greater differences in the thermal expansion coefficients in comparison to the metal. Furthermore, polyamide has an increased moisture content after the climate test and thus lower strength and stiffness. In this work, all polyamide samples were brought to the same moisture content before testing by conditioning according to ISO 1110, while [25] makes no statement on this. Based on the results in this study, a good resistance of the joints against hygrothermal cyclic loading can be stated.

Sufficient wetting of the coating surface with polymer is crucial for the formation of specific adhesion as well as mechanical interlocking and thus a high joint strength. Figure 7 shows cross-sectional images of the interfaces of the spot-joined PA6GF47BD and PPGF47BD specimens. In both cases, almost complete wetting was achieved. The surface microstructures of the coating were almost completely surrounded by polymer. Furthermore, no delamination due to thermal contraction was observed. PPGF47BD showed slightly more strength-reducing cavities than PA6GF47BD, but this was still in a reasonable amount. Thus, the quality of the joints is concluded to be high for both polymers.

In general, a cohesive-adhesive failure between coating and polymer occurred for all tested lap shear specimens. Figure 8 shows the fracture surfaces of a 6L/PA6GF47BD spot-joined specimen without hygrothermal cyclic loading. At the coating side, a high amount of polymer residues (Figure 8a) as well as partially ripped out glass fibers (Figure 8b) were visible, showing that a large proportion of the interface between coating and polymer was able to withstand the occurring shear forces. Moreover, the coating showed no areas of delamination from the aluminum substrate, indicating a strong metallurgical bonding. At the polymer side, a low amount of cavities is visible (Figure 8c). Furthermore, small pieces of the coating adhere to the polymer (Figure 8d). Cohesive fractures in the coating is caused from the brittleness and reduced strength of coated material in comparison to bulk material [28] as well as partially filigree-formed roughness features that are not able to withstand the occurring shear forces.

## 4. Conclusions

In the present work, a technology combination of thermal spraying and inductive thermal joining was developed with the aim of rapidly joining aluminum to thermoplastic polymer. The aluminum was coated with nickel–aluminum 95/5 by wire arc spraying. The coating afterwards served as susceptor for inductive heating as well as adhesion promoter between metal and polymer. Based on the experimental results and the performed analyses, the following conclusions can be drawn:

Nickel–aluminum 95/5 thermal spray coatings offer a rough and undercut surface, suitable for strong mechanical interlocking adhesion to thermoplastic polyamide 6 and polypropylene.Variations in the current and voltage parameters for the thermal spraying process had a large impact on the coating thickness. A decrease in voltage and an increase in current increased the coating thickness. The increase in coating thickness slightly increased the surface roughness values, but led only to a marginal change in the bonding strength between coating and polymer.A different number of coating layers does not significantly affect the bonding strength between the coated metal and polymer adherents.Depending on the coil distance and coating thickness, the coating accelerated the inductive heating process up to 700% in comparison to uncoated aluminum. An increase in coating thickness up to 170 µm leaded to higher heating rates. Since the magnetic field is completely absorbed above a certain coating thickness, a further increase did not lead to a decrease in joining time.Almost complete wetting of the coating surface with polymer was achieved with the spot-joining process.Hygrothermal cyclic loading between 90 °C at 80% humidity and −30 °C at ambient humidity did not significantly affect the bond strength. Thus, a good hygrothermal stability of the joints can be stated.Cohesive-adhesive failure with partially ripped-out glass fibers occurred due to the conducted lap shear tests. The coating showed no delamination from the aluminum substrate.

In future work, the developed technology combination will be transferred into an industrial application. For this purpose, a welding tool for inductive spot-like joining was developed. The tool can be attached to industrial robots and thus automatically join structural components made up from metal and thermoplastic polymer.

## Figures and Tables

**Figure 1 polymers-13-01320-f001:**
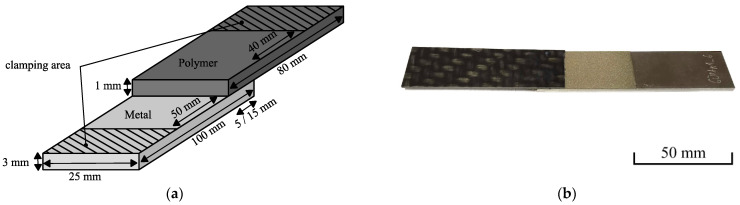
Lap shear specimen (**a**) geometry, (**b**) spot-joined specimen with PA6GF47BD.

**Figure 2 polymers-13-01320-f002:**
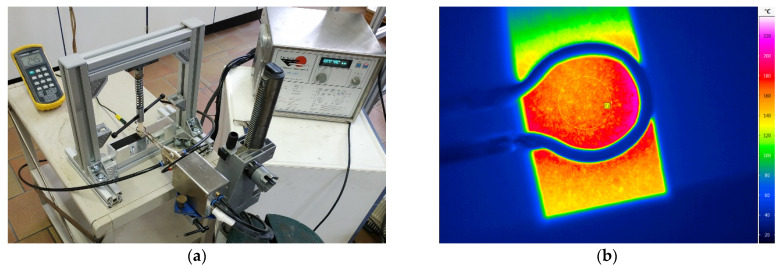
(**a**) Inductive joining device, (**b**) infrared camera image.

**Figure 3 polymers-13-01320-f003:**
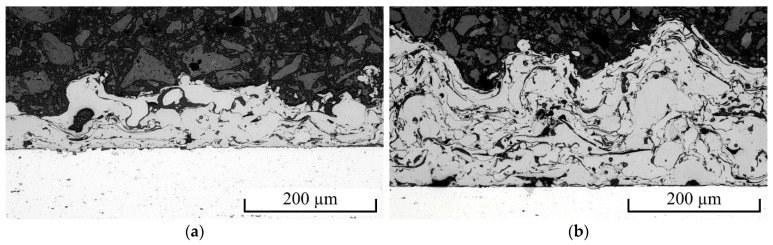
Cross-sectional images of the NiAl5 coatings on EN AW-6082 substrate (**a**) V4 and (**b**) V3.

**Figure 4 polymers-13-01320-f004:**
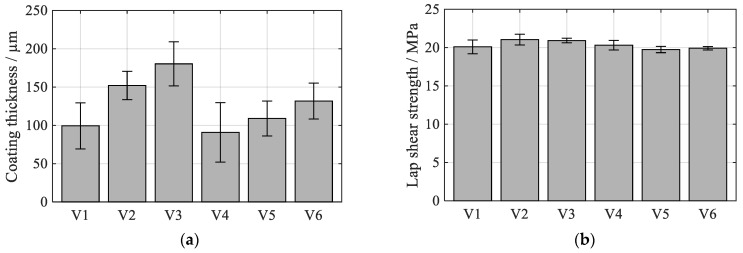
Results for the investigated electric spraying parameters on (**a**) coating thickness and (**b**) lap shear strength to PA6GF47BD; mean values ± 1 SD.

**Figure 5 polymers-13-01320-f005:**
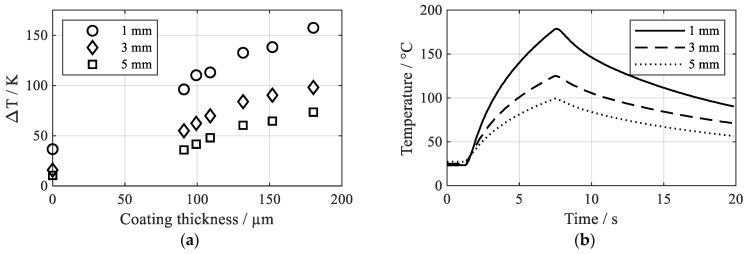
Heating behavior of the coatings, high frequency converter at 50% power setting (**a**) temperature increase after 5 s heating time in dependence of coating thickness and coil distance, uncoated aluminum and parameter sets V1–V6 and (**b**) temperature–time charts in dependence of the coating thickness for the coating parameter set V2 (coating thickness 151.9 µm).

**Figure 6 polymers-13-01320-f006:**
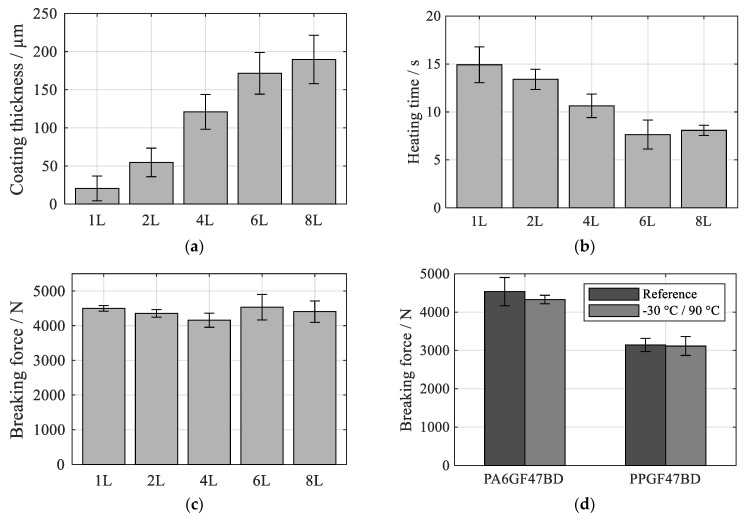
Results for spraying parameter set V2 with 1–8 deposited layers on (**a**) coating thickness, (**b**) heating time for PA6GF47BD at 75% power setting, (**c**) breaking force for PA6GF47BD and (**d**) breaking force for 6L of PA6GF47BD and PPGF47BD before and after climate test; mean values ± 1 SD.

**Figure 7 polymers-13-01320-f007:**
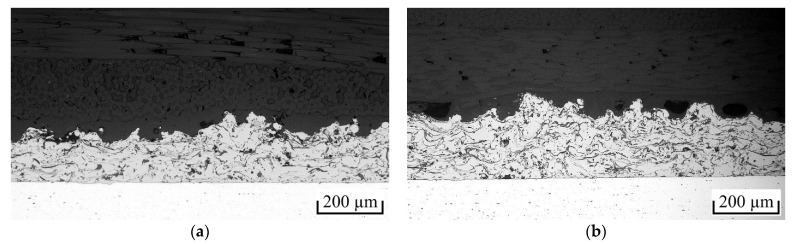
Interface of spot-joined specimens with coating parameter set V2/6L (**a**) PA6GF47BD and (**b**) PPGF47BD.

**Figure 8 polymers-13-01320-f008:**
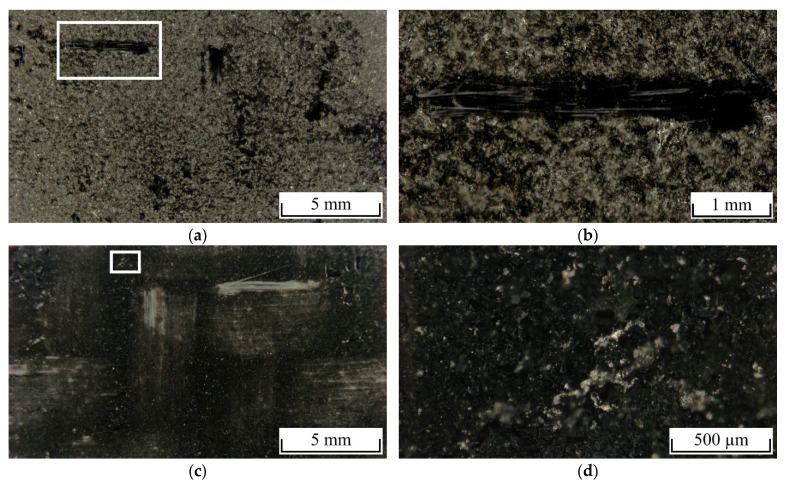
Fracture surfaces of PA6GF47BD specimen (**a,b**) coating side and (**c,d**) polymer side.

**Table 1 polymers-13-01320-t001:** Material properties of the used metal and polymer, supplier specifications.

	EN AW-6082 T6	PA6GF47BD (ISO 1110)	PPGF47BD
Density (kg/m^3^)	2700	1800	1680
Tensile modulus (MPa)	70,000	18,000	20,000
Poisson’s ratio (-)	0.34	-	-
Yield strength (MPa)	240–320	-	-
Ultimate strength (MPa)	300–350	380	430
Elongation to failure (%)	8–14	2.3	2.7
Melting temperature (°C)	660	220	165
Thermal expansion coefficient (10^−6^/K)	23.4	18	11
Fiber	-	E-Glass	E-Glass
Weaving Style	-	Twill 2/2	Twill 2/2
Fiber content (vol.-%)	-	47	47
Thickness per layer (mm)	-	0.5	0.5

**Table 2 polymers-13-01320-t002:** Wire arc spraying parameters for NiAl5 feedstock materials.

Variant	Current (A)	Voltage (V)	Atomising Gas (bar)	Spraying Distance (mm)	Offset (mm)	Feed Speed (m/s)	Layers
V1	120	25	3.5	130	5	1	4
V2	160	25	3.5	130	5	1	4
V3	200	25	3.5	130	5	1	4
V4	120	30	3.5	130	5	1	4
V5	160	30	3.5	130	5	1	4
V6	200	30	3.5	130	5	1	4

## Data Availability

The data presented in this study are available on request from the corresponding author.

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
