# Peer review of "Nickel-Aluminum Thermal Spray Coatings as Adhesion Promoter and Susceptor for Inductively Joined Polymer-Metal Hybrids"

_polymers, 2021, doi:10.3390/polym13081320_

Round 1

Reviewer 1 Report

Thermoplastic polymer-metal-hybrids offer great potential for automotive and aviation applications due to their high strength/stiffness-to-weight ratio. In contrast to hybrid structures containing thermoset polymers, no time-consuming curing process is necessary. Thereby, key challenges are the development of a well adhering interface as well as a fast joining technique. Hybrid joints of metal and fiber-reinforced-polymer offer great potential for lightweight applications. Thereby, a fast and reliable joining process is mandatory for mass-production applications. In this article, this study assesses inductive spot-joining in combination with prior thermal spray coating of the metal adherent. A nickel-aluminum 95/5 coating was applied to achieve high adhesion through mechanical interlocking and to act as susceptor for the inductive joining process. The joint strength was assessed with lap shear specimens consisting of EN AW-6082 aluminum alloy and glass fiber reinforced polyamide 6 or polypropylene, respectively. The joints were further investigated in terms of heating time and hygrothermal cyclic loading. This manuscript should be rejected for published in Polymers. However, if the authors are willing to make the substantial revisions according to my comments, I would be glad to re-review this manuscript. Here are my detailed comments:

  1. The whole introduction has presented a number of related works as literature review. However, the authors simply repeat what those works have done, instead of introducing the brief feature and the coherence of them. Why their work is important comparing to previous reports? I think this is essential to keep the interest of the reader.
  2. Materials and Methods part. Although the results look “making sense”, the current form reads like a simple lab report. The authors should dig deeper in the results by presenting some in-depth discussion.
  3. In Fig. 5, the authors should give the explanations for the difference of data collected from different sources.
  4. It is suggested to discuss what the main advantages the proposed performed analyses has.
  5. Depending on coil distance and coating thickness, the coating accelerated the inductive heating process up to 700%. Hygrothermal cyclic loading between 90°C at 80% humidity and -30°C at ambient humidity did not significantly affect the bond strength. The authors should give some explanation on above conclusions and data.
  6. In order to verify the validity of current results, authors need to compare their results with reported experimental data and models in literatures.
  7. Fiber Reinforced Polymer Composite have been widely used in the industry. In this paper, nickel-aluminum 95/5 thermal spray coatings offer a rough and undercut surface, suitable for strong mechanical interlocking adhesion to thermoplastic polyamide 6 and polypropylene. Actually, surfaces of fiber Reinforced Polymer Composite are rough and have a large influence on mechanical properties of fiber Reinforced Polymer Composite. The surface roughness is an important property of fiber Reinforced Polymer Composite. And several investigators have studied the effect of surface roughness on mechanical properties of fiber Reinforced Polymer Composite, (see [A fractal model for capillary flow through a single tortuous capillary with roughened surfaces in fibrous porous media, Fractals, 2021, 29(1):2150017; Fractals, 2019, 27(7): 1950116 ]). Authors should introduce some related knowledge to readers. I think this is essential to keep the interest of the reader.
  8. Please, expand the conclusions in relation to the specific goals and the future work.

Author Response

  1. The whole introduction has presented a number of related works as literature review. However, the authors simply repeat what those works have done, instead of introducing the brief feature and the coherence of them. Why their work is important comparing to previous reports? I think this is essential to keep the interest of the reader.
    • The benefits of inductive joining in comparison to other thermal joining methods are now explained.
    • The benefits of using a thermal spray coating as susceptor and adhesion promoter were highlighted.
    • The description of the conducted experiments and the findings gained from that were added.
  2. Materials and Methods part. Although the results look “making sense”, the current form reads like a simple lab report. The authors should dig deeper in the results by presenting some in-depth discussion.
    • The discussion was enhanced for the obtained strength values, the heating rate analysis, the climate test results, the wetting analysis and the fracture analysis.
  3. In Fig. 5, the authors should give the explanations for the difference of data collected from different sources.
    • The figure caption was adjusted to provide more details on the depicted charts.
  4. It is suggested to discuss what the main advantages the proposed performed analyses has.
    • The explanation within the introduction, which test methods are used for which purpose, was adjusted to improve the readers understanding. It is now highlighted that the performed coating-related tests indicate the advantage of combining high-strength material connection with a faster inductive joining process.
  5. Depending on coil distance and coating thickness, the coating accelerated the inductive heating process up to 700%. Hygrothermal cyclic loading between 90°C at 80% humidity and -30°C at ambient humidity did not significantly affect the bond strength. The authors should give some explanation on above conclusions and data.
    • An explanation and a reference to reported data for similar experiments has been added.
  6. In order to verify the validity of current results, authors need to compare their results with reported experimental data and models in literatures.
    • A comparison of results to literature findings was added for the obtained strength values, the climate test results and the heating rate analysis.
  7. Fiber Reinforced Polymer Composite have been widely used in the industry. In this paper, nickel-aluminum 95/5 thermal spray coatings offer a rough and undercut surface, suitable for strong mechanical interlocking adhesion to thermoplastic polyamide 6 and polypropylene. Actually, surfaces of fiber Reinforced Polymer Composite are rough and have a large influence on mechanical properties of fiber Reinforced Polymer Composite. The surface roughness is an important property of fiber Reinforced Polymer Composite. And several investigators have studied the effect of surface roughness on mechanical properties of fiber Reinforced Polymer Composite, (see [A fractal model for capillary flow through a single tortuous capillary with roughened surfaces in fibrous porous media, Fractals, 2021, 29(1):2150017; Fractals, 2019, 27(7): 1950116 ]). Authors should introduce some related knowledge to readers. I think this is essential to keep the interest of the reader.
    • We think we can follow the reviewer’s intention, that the used high melt flow index polyamide needs to penetrate the surface roughness features of the thermal spray coating. However, we have not focused on this aspect in our investigations and we would therefore not relate to this paper. Our results (see figure 7) show, that penetration of the polyamide into the roughness features of the thermal spray coating is achieved well through the performed inductive joining processs.
  8. Please, expand the conclusions in relation to the specific goals and the future work.
    • The conclusion was expanded with the aims of the study as well as an outlook to future work.

Reviewer 2 Report

The article 'Nickel-aluminum thermal spray coatings as adhesion promoter and susceptor for inductively joined polymer-metal-hybrids' reports the obtaining of metal-polymer hybrids via inductive joining of EN AW-6082 aluminum alloy to glass fiber reinforced polypropylene and polyamide 6, with the assessment of a self-adhering Ni-Al 95/5 coating.

The authors have amply demonstrated high efficiency and reliability of the method proposed, the article has been properly completed, the main findings and conclusions have been confirmed by correct experimental data.

This work meets the level of the best MDPI journals by the criteria of scientific novelty and relevance, however, in my opinion, the manuscript does not fully comply with Scope of the 'Polymers' journal. Might it not be a good idea to consider possibility of the publishing of this manuscript in more appropriate journal, for example, 'Materials'?

I do not have any substantial criticisms of the paper. The only thing that need fixing is a list of references which should be presented in accordance with 'Polymers' (and other MDPI journals') template, i.e.: Author 1, A.B.; Author 2, C.D. Title of the article. Abbreviated Journal Name Year, Volume, page range.

Author Response

Thank you for reviewing our paper. The changes based on your review are marked in blue.

This work meets the level of the best MDPI journals by the criteria of scientific novelty and relevance, however, in my opinion, the manuscript does not fully comply with Scope of the 'Polymers' journal. Might it not be a good idea to consider possibility of the publishing of this manuscript in more appropriate journal, for example, 'Materials'?

  • In principle, we agree with you that the article is not fully in line with the focus of Polymers, as it has a strong interdisciplinary character. However, this special issue refers to polymer hybrid materials, which we consider in the article. From that perspective, we think Polymers is also appropriate.

I do not have any substantial criticisms of the paper. The only thing that need fixing is a list of references which should be presented in accordance with 'Polymers' (and other MDPI journals') template, i.e.: Author 1, A.B.; Author 2, C.D. Title of the article. Abbreviated Journal Name Year, Volume, page range.

  • The reference list is now presented in accordance with the MDPI guidelines.

Reviewer 3 Report

The main objective of these studies is to develop a new technique for the joining of hybrid composite joints and exploit their importance. The topic is important, the results are interesting and the methodology followed is appropriate, while the content falls well within the scope of this Journal. The overall organization of the manuscript follows the journal layout and recommendation, and it is well organized in a systematic way. Besides the following points required attentions!

  1. The introduction section can be improved by addressing the latest works relevant to the hybrid joint joining techniques and adding some more references to them.
  2. In the materials section, the lack of standards used for the materials have been taken for the specimen preparation, and references are missing for the material properties. Polyamide 6 melting temperature should be corrected with 295 áµ’
  3. Figure 1b is not mentioned in the text,
  4. Lack of standards used for shear strength test and the reasons are not explained why the shear strength variation is very small (almost similar in all), irrespective of the coating thickness and author has not explained.
  5. The explanations for figure 7 and figure 8 are more general, the authors should explain in a more scientific way by comparing the previous results like how does the interface of joint specimens affect the strength and other properties.
  6. In conclusion, the authors mentioned that a variation in electric coating parameters (current and voltage) did not significantly affect the bonding strength to the polymer but the thickness of the coating is not true because at the time of joining it is impossible to form without residues. It can affect strength.

Author Response

Thank you for reviewing our paper. The changes based on your review are marked in green.

  1. The introduction section can be improved by addressing the latest works relevant to the hybrid joint joining techniques and adding some more references to them.
    • Three more references from 2020 and 2021 were added.
  2. In the materials section, the lack of standards used for the materials have been taken for the specimen preparation, and references are missing for the material properties. Polyamide 6 melting temperature should be corrected with 295 áµ’
    • The standards for specimen preparation and testing have been added. The material properties are taken from the suppliers specifications. A note on this was added. The suppliers specification for the melting temperatur is 220°C.
  3. Figure 1b is not mentioned in the text.
    • Figure 1b is now referenced in the text.
  4. Lack of standards used for shear strength test and the reasons are not explained why the shear strength variation is very small (almost similar in all), irrespective of the coating thickness and author has not explained.
    • The used standard as well as the explanation for the small variation in lap shear strength has been added.
  5. The explanations for figure 7 and figure 8 are more general, the authors should explain in a more scientific way by comparing the previous results like how does the interface of joint specimens affect the strength and other properties.
    • The explanations for figure 7 and figure 8 have been enhanced in order to give a better understanding, how wetting behavior affect the joint strength and why fracture takes place in the way shown.
  6. In conclusion, the authors mentioned that a variation in electric coating parameters (current and voltage) did not significantly affect the bonding strength to the polymer but the thickness of the coating is not true because at the time of joining it is impossible to form without residues. It can affect strength.
  •  
    • I am sorry, but I do not understand this comment. I have rephrased the sentence to make it more understandable.

Round 2

Reviewer 1 Report

It is ok.

Reviewer 3 Report

You answered to my question.